# Symptoms of PTSD and Depression among Central American Immigrant Youth

Ernesto Castañeda [1,2,3,*] , Daniel Jenks [1,2,3], Jessica Chaikof [1,3], Carina Cione [1,3], SteVon Felton [1,3], Isabella Goris [1,3], Lesley Buck [4] and Eric Hershberg [2]

1   Department of Sociology, American University, Washington, DC 20016, USA; dj9316a@american.edu (D.J.); jc2539a@american.edu (J.C.); cc1767a@american.edu (C.C.); sf1610a@american.edu (S.F.); ig9220a@american.edu (I.G.)
2   Center for Latin American and Latino Studies, American University, Washington, DC 20016, USA; hershber@american.edu
3   Immigration Lab, American University, Washington, DC 20016, USA
4   Independent Researcher, Arlington, VA 22204, USA; lesleybuck@gmail.com
*   Correspondence: ernesto@american.edu

**Abstract:** The aim of this paper is to explore the mental health challenges that Central American immigrant youth face before and after arriving in the United States. This population is hard to reach, marginalized, and disproportionately exposed to trauma from a young age. This paper investigates the mental health stressors experienced by Central American immigrant youth and asylum seekers, including unaccompanied minors, surveyed in the U.S. in 2017. This mixed methods study uses qualitative data from interviews along with close-ended questions and the validated PHQ-8 Questionnaire and the Child PTSD Symptom Scale (CPSS). These new migrants face numerous challenges to mental health, increased psychopathological risk exacerbated by high levels of violence and low state-capacity in their countries of origin, restrictive immigration policies, the fear of deportation for themselves and their family members, and the pressure to integrate once in the U.S. We find that Central American youth have seen improvements in their self-reported mental health after migrating to the U.S., but remain at risk of further trauma exposure, depression, and PTSD. We find that they exhibit a disproportionate likelihood of having lived through traumatizing experiences that put them at higher risk for psychological distress and disorders that may create obstacles to integration. These can, in turn, create new stressors that exacerbate PTSD, depression, and anxiety. These conditions can be minimized through programs that aid immigrant integration and mental health.

**Keywords:** trauma; unaccompanied minors; migration; Central American youth; transnational families; generational trauma; immigrant integration; social determinants of mental health



## 1. Introduction

Central American youth emigrating to escape violence, poverty, family separation, and abuse may have a high risk for developing mental health issues such as anxiety, depression, and post-traumatic stress disorder. Upon settling in their new community, immigrants grapple with cultural, social, and economic challenges. They must find housing, learn a new language, earn money, navigate prejudice, and establish a social support network. They are thrown into an acculturation process, defined by Alegría et al. as "the acquisition of the cultural elements of the dominant society," which include norms, values, ideas, and behaviors [1]. The relative success of this acculturation process, in turn, shapes immigrants' mental health, their ability to integrate, and how they are perceived by their new society [1]. During the transition, one may accumulate acculturative stress, or the stress of trying to integrate into a new country's culture while seeking to retain one's values, traditions, and beliefs derived from home country experiences [2]. This unique type of stress that

immigrants face upon their arrival can be more difficult to manage by those who have already experienced trauma. Post-migration stressors may inhibit immigrants' recovery from pre-migration trauma, which prolongs and worsens mental health problems [3]. Therefore, the challenges that Central American immigrants face prior to and after arriving in the U.S. will shape their mental health status, vulnerability to further trauma, and integration experience.

Immigrant youth experience major stressors and traumas in their country of origin. Families often separate for economic reasons, having their children stay with a relative while the parents live and work in the U.S. The separation of the family across physical borders is defined as a transnational household [4–6]. Many women immigrants take care of other children abroad while being away from their own children [7]. Despite the distance, many parents still attempt to fulfill familial obligations through remittances: the money and gifts sent back to support their children and express their love and sacrifice [4]. Even though the parents may become stable providers of money, clothes, food, and toys, children are still often unable to comprehend parental separation as anything other than abandonment [8]. Even though children may eventually reunite with their parents in the U.S., feelings of resentment and abandonment may linger [4,8]. Despite the best intentions of the parents, the children left behind often struggle to form close and trusting relationships in their adult lives, including with partners and children [4]. The experience of long-term separation, hardships in their country of origin, and their own migration journey will weigh on their mental health status post-migration. Taking a sociological approach to the social determinants of health [3,9], this paper argues that the migration and integration processes present unique stressors that mold mental health outcomes.

## 1.1. Traumatic Stressors

Driving factors for emigration from Central America include political instability, crime, violence, and low state capacity that deprives populations of access to education, health care, and other services. Migration is one mode of escaping from these hardships. Trauma is defined as "actual or threatened death, serious injury, or sexual violence" [10]. Surviving a trauma is not only stressful, but it may create long-term sequelae, and in some cases, develop into a chronic stressor such as PTSD. Immigrants may experience trauma at different points of their migration process: pre-migration traumas in their home country, traumas on the migratory journey, and a hostile environment in the new country [11]. Common forms of pre-migration trauma include natural disasters, war, gang violence, victimization, witnessing a crime, physical and sexual abuse, or attacks based on their sexual orientation or gender identity. Migrant youth traveling alone face an increased risk of undernutrition, dehydration, assault, kidnapping, and others forms of violence [12–15]. These traumatic experiences put migrants at higher risk for psychological distress and disorders that may also create obstacles to integration that create new stressors that accumulate into compounded traumas [11].

There are different pathways that shape the lives of migrant families and their children. While qualified workers, refugees, and asylum seekers are often allowed to immigrate as family units, most often families are separated as a result of migration. For example: immigrating without a child due to limited options for legal migration, forced separation that occurs either after arriving or during their time in the U.S., and the migration by unaccompanied minors. Often, migrants must make the journey to the U.S. without their children so as not to expose them to the perils of traveling or because they plan to work abroad temporarily to send remittances [16]. Other families are forcibly separated upon arriving at the U.S. border or after they have made it across and established their new lives. These parents who leave their children behind are operating under structurally constrained choices and are forced to sacrifice present needs for their children's future economic security [17,18]. When limited options force families to separate, their decision is contingent upon what the family believes to be most beneficial in the long run [19]. Parents

may view immigrating to the United States as a means to achieve economic stability for the family [19]. Nonetheless, children left behind may feel abandoned and resentful [4,6,20].

Often looking to reunite with parents, minors who arrive at the border without any adult family members are designated by U.S. authorities as "unaccompanied alien children" (UACs) and are subject to additional legal protections. However, legal representation is not guaranteed, and the "fast-track court hearings" [11] that follow their arrival limit opportunities to develop legal claims for immigrant youth. Once in the United States, immigrant children face additional risks for mental health disorders due to poor living conditions, lack of opportunities, discrimination, limited access to federal resources, and fear of deportation [11].

Children from families divided across borders have a higher likelihood of experiencing separation anxiety, ongoing grief, and low self-worth [4,8]. Familial cohesion and stability play a role in behavioral outcomes among immigrant youth [21]. Those who are cared for by their parents or relatives have better behavioral outcomes than those who do not experience familial supervision or guidance [21]. In addition, a caregiver's documentation status affects the well-being of the youth in their care. Immigrant youth who live with undocumented caregivers are more likely to be stressed [21].

Young immigrants may have a heightened vulnerability to post-traumatic stress disorder (PTSD) [22]. Asylum seekers and immigrant youth may have higher rates of PTSD [23–26]. Immigrant youth may be at an increased risk for PTSD due to (1) pre-migration trauma; (2) traumatic migration experiences; (3) migrating unaccompanied; (4) experiencing higher acculturation stress if migrating at an older age; (5) experiencing prolonged family separation; (6) threats of deportation and forced separation; and (7) discrimination and hate crimes [27]. Furthermore, they may struggle to maintain healthy adult relationships in the future [4]. In this way, migration can be understood as a social determinant of mental health among immigrant youth and adults.

The mental health of immigrants is not only connected to the trauma they face prior to arriving in the U.S. but also to U.S. government policies, which have a long history of excluding Latin people (we use Latin as a gender-neutral adjective, we also use Hispanic as synonymous, when used by the authors cited) [28], and which have become increasingly restrictive since the 1990s. Immigrants and their children often face a hostile and xenophobic social and political environment. The changes in policy over the past three decades primarily targeted undocumented immigrants who had previously managed to fly under the radar [29]. The Illegal Immigrant Reform and Immigration Responsibility Act of 1996 (IIRIRA) established a "bar of inadmissibility" for five to ten years for undocumented immigrants who overstayed their visas and allowed for deportation without counsel or legal representation [11,29]. Additionally, the IIRIRA increased resources for immigration enforcement agencies such as Customs and Border Patrol. The IIRIRA was not the only discriminatory policy passed at this time. The Personal Responsibility and Work Opportunity Reconciliation Act of 1996 (PRWORA) restricted undocumented immigrants' rights to social services, including access to food stamps, healthcare, and Social Security [29]. These acts removed the federal government's responsibility to grant aid to immigrants and allowed state governments to limit or exclude immigrants from federal and state programs with the belief that they had not been in the United States long enough to be entitled to such services [29]. These policies played a critical role in subjecting immigrant children to a higher risk of mental health disorders that included PTSD, anxiety, and depression by restricting access to mental health care.

In addition to IIRIRA and PRWORA, state laws such as Arizona's Support Our Law Enforcement and Safe Neighborhoods Act (SB 1070) subjected immigrants to additional perils. SB 1070 gave law enforcement agencies the ability to detain anyone who was suspected of being an undocumented immigrant. Anyone who was not carrying a legal residency document would be charged with a misdemeanor. The primary goal of SB 1070 was to reduce the number of undocumented immigrants in the U.S., to encourage self-deportation, and to discourage new immigrants from entering the country. In practice,

this law criminalizes people simply for appearing to be Hispanic and thus assumed to be undocumented [30]. Another study showed the unfavorable impacts of racial profiling on high school-age Hispanics, who showed a higher probability than Black and white students to report feeling sad. Hispanic students were also slightly more likely to have suicidal thoughts and attempts than others [31].

In this way, pre-existing traumas and subsequent mental health struggles influence immigrants' psychological well-being and integration experience in the U.S. A different study reported that female immigrants from Latin America experienced high levels of trauma from domestic, community, emotional, physical, and sexual violence. Some girls experienced abuse at the hands of relatives with whom their parents had left them after migrating. A minor could be subjected to emotional violence by being left behind and forced to deal with neglectful or abusive caretakers. In addition to experiencing childhood violence, women dealt with physical violence from their domestic partners. Sometimes, physical violence was considered normal, and family members encouraged women to stay in the relationship [32]. Furthermore, sexual abuse is underreported by males [33].

This is not to say that every migrant goes through traumatic experiences. For example, some immigrants make the journey by plane and enter the U.S. with immigrant visa or as tourists, so they experience less violence and trauma on the journey. However, those without prior legal recourse to family reunification often complete the journey over land, becoming exposed to life-threatening incidents with gangs, thieves, or human smugglers [32]. After arriving in the U.S., many experience sexual and physical violence but feel they cannot report it to authorities due to their immigration status [32]. This shows the mental health advantages of providing administrative avenues for immigration and family reunification, as it would make migration a safer experience for all. Racist immigration policies that increase animosity against immigrants have heightened the fear of deportation and worsened immigrants' mental health [11,28].

### 1.2. Mental Health among Immigrants

After arriving in the United States, many immigrants experience challenges associated with language, economic hardships, and discrimination. The demands created by mainstream society and the lack of social support can contribute further to acculturative stress [1,34,35], migration-related stressors [36,37], and psychiatric disorders [35]. A study by Finch and Vega found that as someone born outside the United States acclimates to their new home, their perception of discrimination slowly increases as they learn English and become familiarized with their environs [38]. This increased awareness of discrimination and acculturation stress led the researchers to find a significant correlation between acculturation and depression. With Latin youth populations being at the highest risk for depression among multiple ethnic groups [39], it is not surprising that the perception of discrimination among migrants leads to a significant decrease in self-esteem [40]. In multiethnic states like Florida and California, 55% of adolescent Latin individuals have experienced at least one form of discrimination [40]. Central American immigrants with high levels of acculturative stress are more likely to experience depression, suicidal ideation, and anxiety [41,42]. Hovey concluded that this might be because they feel caught between two different cultures [34]. However, Dunn and O'Brien report that Latin immigrants feel the pressure to learn English quickly but experience lower stress levels assimilating into American society compared to other groups [43].

Factors that protect immigrants from mental health disorders, such as family cohesion, may erode over time due to the labor market demands and the challenges of having to negotiate between two cultures. According to Cook et al., Latin immigrants, who have lived in the United States for less than ten years, experience lower rates of psychiatric and depressive disorders. Conversely, Latin immigrants living in the U.S. for over twenty-one years are more likely to have a psychiatric disorder than their U.S.-born counterparts [35].

Recently arrived immigrants often lack opportunities to secure medical insurance and therefore cannot access mental health services. In addition to lack of insurance, other

barriers to accessing mental healthcare are distrust of the medical system, lack of childcare or transportation, and unfamiliarity with mental healthcare. In the United States, only some immigrants access mental health services to potentially mediate or lessen symptoms of PTSD, depression, and anxiety. Only about a third of Latina immigrants referred to mental health services receive therapy or psychiatric consultation; younger Latinas and those experiencing only anxiety use mental health services less frequently [44]. Immigrants who experience longer elapsed time between the referral and the intake appointment are less likely to use such services. Houchasen and co-authors argue that this may be due to the lack of providers, the unavailability of appointments, clients not understanding or agreeing to referrals, although case managers are key in helping immigrants receive professional mental healthcare [44].

## 2. Materials and Methods

The project from which this paper derives, "Household Contexts and School Integration of Resettled Migrant Youth," included: semi-structured interviews with 23 social service providers, school staff, and community experts in the metropolitan Washington, D.C. region; a structured interview with 41 parents or sponsors of minors who arrived in the region; as well as a structured interview with closed- and open-ended questions and two mental health symptom scales applied to 58 Central American participants who arrived in the United States as minors a few years before the time of the interview (2017). We conducted interviews with sponsors and experts to triangulate our qualitative findings. For reasons of length, in this paper, we mainly report the findings directly from youth respondents; however, many of the youth statements were further substantiated by some of the sponsors and service providers.

Unaccompanied minors from Central America have been a frequent topic of discussion in the news since at least the so-called border crisis of 2014 [13,45], but with few exceptions, their voices are rarely heard. These exceptions include a few journalistic snippets, *testimonios* (first-person accounts) [46], interviews of detainees [47,48], or consideration of art they produce during detention [49]. Children have agency and their own way of processing and understanding their experiences and conditions [17], and the best way to understand their mental health outside of a clinical setting is through interviewing and assessing them directly.

Unaccompanied minors and recently arrived immigrant youth constitute a subpopulation that is vulnerable and hard to reach. Therefore, our sampling methods included recruitment through after-school programs, advocacy and legal services, non-profit agencies, and snowball sampling. Participants were paid $25 USD for their participation. Interviewers were certified in human subject ethics training and trained in trauma-informed interviewing. The Institutional Review Board of American University approved the study [IRB-2016-227]. The PIs had previous experience working with undocumented, homeless, disabled, and vulnerable populations. We made clear that we were not working for the government and were not lawyers or journalists. All the PIs and research assistants spoke Spanish. Many of the interviewers were Hispanic and some Central American. They were often immigrants or children of immigrants. Most interviewers were female. The interviewing team included undergraduates and masters level students, so establishing rapport with the interviewees was relatively easy, and it was possible to gain a certain level of trust and confidence.

The survey instrument gathered demographic information as well as details regarding dangers in the youths' home countries, migration journeys, family dynamics, U.S. sponsors, and the youths' experiences with immigration courts and schools in the United States. Youth were also asked whether they had spent time in a detention center, shelter, health clinic, or rehabilitation center over the past six months. Self-reported mental health status, PTSD symptoms, and depressive symptoms were assessed using the Child PTSD Symptom Scale (CPSS) and the Patient Health Questionnaire (PHQ-8) scales. The interviews were

wide-ranging. Some questions were open-ended, whereas others invited simple yes or no answers, and others entailed scale-ranking.

Mixed method studies involve using different approaches to collect or analyze data to increase confidence in the findings [50–54]. We designed this to be a mixed methods study using closed-ended quantifiable questions and open-ended questions that would elicit narrative responses. The purpose was to triangulate the qualitative and quantitative responses collected simultaneously from the same respondents during the same point in time.

Our goal was to explore the mental health challenges that Central American immigrant youth face before and after arriving in the United States. A mixed methods approach provided us with the opportunity to bring forward the voices of this vulnerable and hard to reach population with confidence, points of comparison, and verification to make up for the sample size. The point of the mixed methods design and asking similar questions in different ways was to enable us not to take any answer at face value but to compare answers given by the same respondent.

On average, interviews lasted between 60 and 90 min. All interviews were conducted in Spanish and transcribed in-house, and the narrative sections selected as representing important trends and cases were translated to use in English-language publications after the coding and analyses were done. This ensured that the text remained faithful to the original meaning. Interviews were recorded, transcribed, entered into Qualtrics to analyze responses to the same answer, and coded in NVivo using spelled-out coding trees to bring together themes brought up by participants at any point during the interview. We analyzed quantitative variables on SPSS, thus allowing for the triangulation of both qualitative data and descriptive quantitative statistics.

Thirty-seven youth reported El Salvador as their country of origin, followed by Honduras and Guatemala with 16 and 5 interviewees, respectively (see Table 1). Age at the time of migration ranged from 8 to 20, with only two participants reporting more than 18 years (19 and 20). We decided to include them because their experiences were very similar to those 14 or 16 who already saw themselves as of working age [55]. Interviews took place when youth were at a minimum of 10 years old and a maximum of 22 years old, and the average age at the time of the interview was 16. At the time of their arrival at the border, 34 were unaccompanied, and 24 were accompanied. Additionally, nine arrived with documentation, and 49 minors were undocumented. When interviewed, 22 participants resided in Prince George's County, Maryland, USA, 24 in Montgomery County, Maryland, USA, and 12 in Fairfax County, Virginia, USA, all of which are core jurisdictions in the Washington, D.C. metropolitan area.

Because the survey was comprehensive and asked questions about minors' lives in their home countries, in the United States, and during their migration journeys, we anticipated that there would be changes in youths' mental and physical health statuses post-migration. To track this, question 8 of the survey asked respondents to "Describe your health status prior to migration" by checking "yes" or "no" to the following mental health-related conditions: "Constant Stress," "Anxiety (unease or excessive concern)," and "Depression." Respondents were given these options in Spanish: "Estrés constante," "Ansiedad (intranquilidad o preocupación excesiva)," and "Tristeza." At the end of the survey, they were asked the same question about their health after migration.

Respondents filled out a PHQ-9 modified for teens in Spanish to screen for depression symptoms minus the questions related to suicidal ideation, also known as PHQ-8 [56,57]. This scale has been validated in English and Spanish and is often used by psychologists and clinicians internationally [58,59]. The instrument asks if respondents had experienced individual depression symptoms in the previous two weeks on the scale of (0) "None" (Ninguno), (1) "Various days" (Varios días), (2) "More than Half" (Mas de la mitad de los días"), or (3) "Almost every day" ("Casi todos los días"). We scored as follows in Table 2.

**Table 1.** Descriptive statistics among immigrant minors in the D.C. metropolitan area (*n* = 58).

| | Overall | |
|---|---|---|
| | *n* | Years/% |
| Average age at time of arrival | 58 | 14 |
| Average age at time of interview | 58 | 16 |
| *Gender* | 58 | |
| Male | 31 | 53.5% |
| Female | 26 | 44.8% |
| Non-binary | 1 | 1.7% |
| *U.S. legal citizenship status at time of arrival* | 58 | |
| Documented | 9 | 15.5% |
| Undocumented | 49 | 84.5% |
| Country of origin | 58 | |
| El Salvador | 37 | 63.8% |
| Honduras | 16 | 27.6% |
| Guatemala | 5 | 8.6% |
| Jurisdiction within DC metropolitan area | 58 | |
| Prince George's County | 22 | 37.9% |
| Montgomery County | 24 | 41.4% |
| Fairfax County | 12 | 20.7% |
| *Accompaniment status at the border* | 58 | |
| Accompanied | 34 | 58.6% |
| Unaccompanied | 24 | 41.4% |

**Table 2.** PHQ-8 scoring guide.

| SCORING | SEVERITY |
|---|---|
| 0–4 | No or minimal depression |
| 5–9 | Mild depression |
| 10–14 | Moderate depression |
| 15–19 | Moderately severe depression |
| 20–24 | Severe depression |

Source: [60].

Respondents filled out a Spanish version of the Child PTSD Symptom Scale (CPSS), a version of the PCL-5 used by the Cognitive Behavioral Intervention for Trauma in Schools (CBITS) designed to screen for PTSD in children and adolescents [61]. The instrument is routinely used to provide services and was translated by the Los Angeles Unified School District. It is a self-reporting checklist assessment utilizing the DSM-4's definition of symptoms apparent in those suffering from post-traumatic stress disorder (PTSD). The CPSS assesses PTSD symptoms and diagnoses in children ranging from ages 8 to 18 based on three items: reexperiencing trauma, avoidance, and arousal [62]. The 17-question assessment asked respondents to report and rate symptoms they experienced in the past two weeks. They were asked to choose how often they experienced symptoms by selecting (0) "Not at all" (Nunca), (1) "Once in a while" (Ocasionalmente), (2) "Half the time" (El 50% del tiempo), or (3) "Almost always" (Prácticamente en todo momento). Following CBITS scoring guidelines, we used a cutoff of 14 points or higher to indicate moderate to severe symptoms of PTSD (see Table 3). This study purposely did not use a checklist of traumatic experiences to avoid triggering study participants, and thus making participation itself traumatic. Respondents were invited to talk about feelings and symptoms, self-report mental health conditions, and self-disclose traumatic experiences, if they felt comfortable.

**Table 3.** Child PTSD Symptom Scale scoring guide.

| | Add the Number of Points Endorsed Per Question, Where |
|---|---|
| 1 | not at all = 0 points, once in a while = 1 point, half the time = 2 points, and almost always = 3 points. |
| 2 | Tally all points for each of the 17 questions to obtain a total score. |
| 3 | Total scores of 14 points or higher indicate moderate to severe PTSD. |

Source: [62].

## 3. Results

### 3.1. Traumatic Experiences

Most of the youth and adolescents we interviewed had previous traumatic experiences. Some youth shared that they had seen loved ones killed in front of them and personally experienced violence and abuse. Here, we present three excerpts of participant narratives as examples of some of the experiences that youth shared. These narratives point to stressors that youth experience before, during, and after migration.

#### 3.1.1. Carlos

Carlos, who was 15 when interviewed, briefly described his life in El Salvador, which often felt indirectly dictated by the gangs:

"They wanted to force me to join the Maras. And that is why you can't study: because I was scared to leave the house . . . to go to school."

Death threats were also common if youths did not want to join the gangs. Carlos continued:

"They only followed me once, but they didn't get me. I headed home. If someone doesn't join the Maras, they kill you, young. There are no options. If you don't join the Maras, they kill you. I felt a lot of pressure."

For Carlos, coming to the United States and joining his father, already living there, was his preferred option over death or joining a gang. He spoke with his family, and then his father arranged for and paid for him to travel north. When asked why he migrated, he responded:

"Fleeing the Maras. It was my idea to come . . . My dad paid so that they [the coyotes] could bring me. From El Salvador, I went to Guatemala; from Guatemala, I took a bus. I got off, took another bus, got off. At night the buses ran . . . I must've taken around twenty buses from Guatemala to Mexico. From Mexico to the United States, I went by bus and by taxi. I crossed the border and was walking when they detained me. They took me to Immigration; they interviewed me. I explained my case, why I came . . . I was at the center for minors for about half a month. I played there, they had classes, and from there I went with my dad. The trip took around three months . . . I was in a migration center, but not the Court. I understood the rules: I shouldn't miss school, I shouldn't work . . . mostly that . . . In the center for minors, they treated me well. In the immigration center, they speak very angrily to you. I felt nervous because they spoke to me very angrily."

Overall, youth understood that they were under strict watch while their cases were being processed. For Carlos, age 15, he had to attend school, not work, and stay out of trouble—anything that he did, he was told could be used against him in immigration proceedings. Carlos said the lawyer, "didn't help us that much . . . I don't know anything regarding the decision on my case."

3.1.2. Diana

Diana's story is emblematic of the stressors that many like her experience. The words of Diana, 16, from El Salvador, echoed those of Carlos about confidence in the lawyers and courtrooms that they had to experience.

"No, we go to the court, and then, they don't tell us anything. The lawyer said that they weren't going to give us anything, that I mean, we didn't have hopes that they would give us, I don't know, a permit [to stay]."

She explained why she wanted to leave El Salvador—she was only 13 when she left.

"I came from El Salvador because I wanted to study more. I want to be a doctor. Where I was living then, the schools aren't great. Well, they don't teach a lot of things, like math, or things like that. They only taught us math twice a week, for 30 min. Same with science. So, my parents wanted me to come here to study, and I also wanted to study, but where I lived, I couldn't study what I wanted to study."

She felt more able to attain those goals in the United States than in El Salvador. She continued:

"Here I think when I'm 22, or when I'm no longer a minor, I'll be going to university. I want to study. And where I used to live was very dangerous, and there are no opportunities to pursue higher education. Most people in my country only study up to high school and stop there. It is very rare that someone makes it to university."

Diana spoke about her scary, confusing, and dislocating experience crossing the border and being kept in Customs and Border Protection (CBP) custody:

"We were in a house, and then a man told us that he was going to leave us in a forest. There was a forest, and you could hear a river. And we were on a rock with some other men, who were also with the man who brought us. And you could hear animal noises, and they were very scary. We were scared. It was around 5 in the morning, and we were there for around two hours waiting for the man who was supposed to guide us across. We passed the river once, but then that was when we were going to walk along in the desert. Then we came back and did another loop because the man made a mistake, and he said that we were going to pass through there to Mexico. So, then we came back to go through the United States, and the man was mistaken again. So, then we came back to Mexico, and then the next time we crossed it was where cotton was being grown, and then we walked a lot, a lot to get to a big, black gate where there was an immigration van waiting, and they detained us. They asked our names, and they loaded us in their car, and took us to a little house where there were only around 15 girls. They gave me a blanket, around, well it must've already been nighttime by then. They took us to shower, because our clothes were all [dirty], and they gave us [new] clothes. The next day they took us to the detention center."

Diana described that at first, she struggled in school and felt like she did not know any of the course content because of inadequate schooling in El Salvador. She was nervous that there would not be anyone she could relate to in school. But she found people to help her academically, and she was able to make friends:

"It was very taxing. At first, I did not want to go out. I was afraid of people who I thought did not speak Spanish. I don't know, I thought they would say go, you aren't from here, things like that. And I didn't want to go to school, I felt sad. But after the first day I went to school, I made friends and then I didn't even want there to be a weekend. I only wanted to be in school. Because we played. And although some classes/subjects were difficult for me, like math, because let me tell you, in El Salvador, they barely had taught me how to add. I didn't know how to multiply, I didn't know how to divide, I didn't know anything! I

only knew how to read, but I didn't read very well. And here they taught me in Spanish. They are teaching me how to read better. And I am interested in school here. They have a lot of programs to help us, for the students at our school."

### 3.1.3. Samantha

Samantha, who was 15 and from Honduras, explained how her group was held up near the Guatemala-Mexico border by gangs with firearms threatening to light her group's buses on fire, with everyone inside, unless they all paid a fee. She had to sleep in a field for four nights while the coyotes arranged with the gang to let them pass:

> "And they went around in cars too, with firearms and all that. They would go in front of cars and wouldn't let them pass. Supposedly, that post is controlled by them. *In which country was this?* Guatemala, and I think it was on the border with Mexico. And if you didn't pay them a certain quantity of money, they wouldn't let you through. Supposedly, they were going to set the buses on fire because they were locked from the outside. I mean, no one from the inside could open the doors or anything. The others couldn't open them or anything. *How was this problem resolved?* I don't know how they resolved it. They, I think they went back to where they came from. We were left at that sports field, that I told you about. We slept and spent around four days there, sleeping in the cold and everything. They made an agreement with them. And they gave them, the people that had to give them money, I believe a week. And if not, well that they'd do away with us. Uh-huh, and I think they were able to get it and afterwards we were let free. And we left."

As Samantha continued, she recounted that she only got to eat food once a day during the trip and would otherwise drink water. Traveling through Mexico was not easy.

Samantha described how she had not seen her mom in years and was very little when she left for the U.S. The first year in the U.S. was hard for Samantha because she missed her grandmother and sister, who had felt like her mom more than anyone else when she was still living in Honduras. Other youth, including Melissa, age 14, David, age 13, and Sarah, age 18, also reported that one of the more challenging parts of coming to the U.S. was leaving other relatives, particularly their grandmothers.

### 3.2. Self-Reported Mental Health

In each mental health "yes" or "no" question in our survey, our respondents by-and-large stated that they did not experience "mental health problems" ("*problemas de salud mental*") before or after migrating to the United States, with only 3 out of 58 responding "yes" (Figure 1). However, many reported having experienced feelings of sadness, worry, and restlessness. About sixty percent of youth reported feeling sad before and after migration. While there was a slight decrease in feelings of sadness post-migration, the change was less than three percent. About one-third of youth reported that they had feelings of anxiety, worry, and restlessness both before and after moving to the United States. Additionally, after immigrating, some reported a decrease in constant stress. This implies that either living in their home countries, making the migration journey, or both situations put a disproportionate amount of stress on the Central American immigrant youth in our sample. Furthermore, many had traumatic experiences. We do not display this data to show population prevalence rates but to report results from our sample and to contrast the mental health self-reported by participant minors with the results from both the scales and the traumatic experiences shared during the same meetings.

The individuals answering in the affirmative for any of these conditions were not always the same pre- and post-migration. There were no statistically significant differences in these self-reported concerns before migrating (through recollection) and at the time of the interview. No statistically significant differences were found based on gender. However, the three individuals noting mental health problems before migrating are all males. The self-reported mental health of our participants slightly improved after migration.

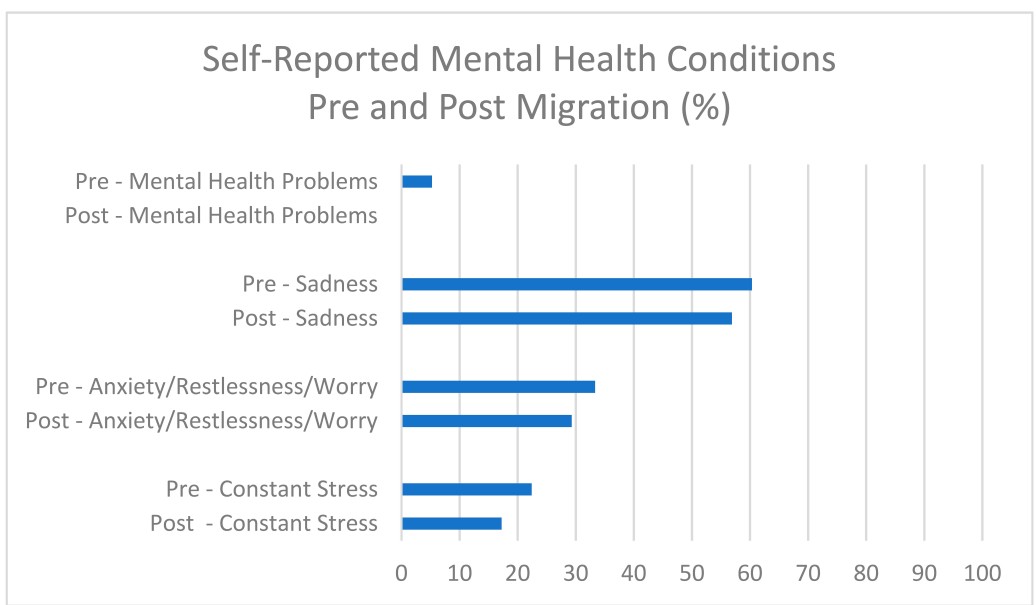

**Figure 1.** Self-reported mental health pre- and post-migration.

### 3.3. Mental Health Conditions Pre- and Post-Migration

In the open-ended and follow-up questions, the participants talked more about their sources of worry before and after migration. Table 4 summarizes the types of responses most often given.

### 3.4. PHQ-9 Modified for Teens

Using the modified PHQ-9 for teens, referred to here as PHQ-8, we were able to examine the severity of depressive symptoms among participants. While 31% of participants reported that they had not struggled with any depressive symptoms within the last two weeks, 79.3% scored between 0 and 4 in the PH8-scale, indicating no or minimal depression, and around 20% exhibited symptoms of mild or moderate depression (Table 5). In one of the follow-up questions, not to be used for scoring, respondents were asked if they felt sad or depressed most days in the last year. Contrary to their other answers, 38% of them responded "yes." We did not ask for suicidal ideation or follow up with a formal diagnosis by a clinician, so the depression scores are on the conservative side.

### 3.5. Child PTSD Symptoms Scale

Based on the Child PTSD Symptom Scale, 66.7% of youth showed low to no symptoms of PTSD. However, 33.3% of youth did show symptoms that could indicate moderate to severe symptoms of PTSD. Although answers to this assessment would not be used as stand-alone criteria to measure or diagnose PTSD, they do allow researchers to assess individuals preliminarily, monitor present symptoms, and examine whether symptoms indicate a PTSD diagnosis.

The results shown in Table 6 are heterogenous: 12.3% reported no symptoms of PTSD; 54% reported a few symptoms; and 33% scored above the cutoff of 14 points. However, 11 participants, of the 47 that answered the scale, scored between 20 and 28. It is noteworthy that two individuals had very high scores, 42, and 45, close to the maximum possible score of 51.

**Table 4.** Mental Health Worries and Stressors Pre- and Post-Migration.

|  |  | Reasons before Migration | Reasons after Migration |
|---|---|---|---|
| Constant stress | | • Want to see and reunite with parent(s)<br>• Gang presence<br>• Living alone<br>• Planning the trip | • Having to stay indoors and being bored<br>• Work |
| Anxiety | | • Future dreams; the trip<br>• Violence affecting friends<br>• Gangs<br>• Needed parent<br>• Could not afford school supplies<br>• Could not study around delinquency<br>• Payment to gangs or coyotes | • Missed and worried about family in home country<br>• New U.S. president<br>• Possible family deportation<br>• Future life<br>• Exploitation at work<br>• Bullying |
| Sadness | | • Leaving friends and family<br>• Could not get out with gangs<br>• Lacking things that others had<br>• Missing parent<br>• Mother: not knowing her, wanting to meet her<br>• Death of grandmother<br>• Death or disappearance of friends<br>• Alcoholic father | • Family separation<br>• Missed friends |
| Mental health problems | | • Frustration<br>• Aggression<br>• Headaches | • None |

**Table 5.** Symptoms of depression from the PHQ-8.

|  | *N* | Percent |
|---|---|---|
| No or minimal | 46 | 79.3 |
| Mild | 8 | 13.8 |
| Moderate | 4 | 6.9 |
| *n* | 58 | 100 |

**Table 6.** Symptoms of PTSD from the Child PTSD Symptom Scale.

|  | *N* | Percent |
|---|---|---|
| Zero | 7 | 12.3 |
| Low | 31 | 54.4 |
| Moderate to Severe | 19 | 33.3 |
| *n* | 57 | 100 |

## 4. Discussion

### 4.1. Traumatic Experiences

Carlos discussed how the gang violence in his home community inhibited his ability to study, move around, or take advantage of opportunities outside of school. Once he was settled in the D.C. area, he said that he felt very safe and that his father kept a close eye on him and his siblings—he felt cared for. Uncertainty of whether Carlos will be allowed to stay with his father in the D.C. area or instead be sent back to El Salvador, where he

has nowhere safe to stay, is a persistent source of anxiety, as is the case with liminal legal status [63,64].

He described having difficulty in school due to language barriers; however, like other youth in our study he was able to develop friendships when he found classmates who were also still learning English. Importantly, now he can imagine doing more than he could when he was in El Salvador. He described not being able to leave the house in El Salvador; now he feels he can join a soccer team in the short term, and in the long term, attend university to become a lawyer. By being taken care of by his father, he was able to start seeing the possibility of a better life for himself [21]. This underscores the importance of family reunification for mental health and the role that family support plays in youth integration.

The trauma of witnessing or being surrounded by violence in migrant-sending communities, family separation, and a notoriously dangerous journey North can last far longer than is clearly visible. For instance, multiple participants reported that one or both of their parents were killed by gangs in their sending communities when they were very young. The other struggles described above, some but not all involving violence, can play into different forms of mental strife as well.

The mental health findings in our sample are mixed. While some mental health measures were positive within the sample, others were negative even with the same individual. One minor, Diana, reported that she wanted to come to the U.S. because she knew that the educational opportunities were better, and she was determined to become a doctor and return to El Salvador. However, she also reported that she sometimes felt isolated and sad. She felt as though her dreams might never be achieved, despite the strong support system of faculty and peers at her school. Another interviewee, Samantha, spoke about how she felt safer in the U.S. than in Honduras. However, she also experienced the loneliness that Diana described, as well as background feelings of tiredness and not having the energy to go about anything beyond the minimum of everyday life. Her mental strife could stem from the fact that she seldom felt safe in Honduras and that she experienced constant fear for her immediate physical safety on the journey from Honduras to the U.S. As demonstrated here, Diana and Samantha both have experienced complex traumatic situations and subsequent feelings of isolation.

These feelings of loneliness may also be due to prior trauma, such as being left behind when their parents migrated to the United States. The parents often have attempted to compensate for this sense of abandonment through "teleparenting" and by sending advice, admonitions, money, clothes, and toys from afar [8]. Yet many children reported feeling ungrateful for what parents understand to have been sacrifices made for the benefit of their offspring. Immigration leaves both the child and the parents sad and uncertain about the future [8].

*4.2. Self-Reported Mental Health*

Figure 1 shows how many of our respondents reported sadness, anxiety, and/or stress as well as traumatic events but did not consider themselves as "having mental health problems." This could be framed as resilience or as a reflection of their understandings about mental illness. However, an important finding from the triangulation is the discrepancy in self-reporting between "mental health problems" and other mental health aspects. Youth may not consider their anxiety, depression, or constant stress to be a mental health problem. Our findings related to symptoms of PTSD also tell us that more of our sample may be experiencing mental health problems but simply not conceptualizing these symptoms as part of a larger whole. One potential reason for this discrepancy may be the widespread stigma around mental health [65,66]. Additionally, adolescents generally have a harder time understanding their emotions and identifying healthy adaptive emotion regulation strategies than adults [67,68]. A lack of emotional awareness, coping skills, and mental health information might account for participants' tendencies to report negative emotions

but not mental health problems. By using the term "mental health problems," the study addresses issues that have not been formally diagnosed by a mental health professional.

Our qualitative survey results also tell us that our sample had many challenges to confront, often throughout their lives. Children and parents often struggle to understand one another after reuniting physically in the United States, there can remain unsaid tension, recurrent feelings of abandonment, and struggles getting to know one another (again or for the first time). Youth reported that getting used to the relationship with their parents or other sponsors was hard, though it improved over time. Often there are new stepfamilies in addition to the blood relatives of the children, and the dwellings that house immigrant families may be small, cramped, and offer little privacy or space.

### 4.3. Mental Health Pre- and Post-Migration

Children are likely to be safer in the United States than they were in their country of origin. However, preoccupations about documentation status and deportation impact their mental health [21,69]. Immigration status can contribute to anxieties about immigration authorities and being sent back to an unsafe environment in their home countries. Children fear making friends because no one will understand their situation or, in a worst-case scenario, notify authorities of their presence [28]. Their feelings of abandonment, the separation from their parents, and the possibility of being turned over to the authorities are all potential contributors to feelings of isolation.

For Central American minors who were forcefully separated from their parents at the border, this experience may contribute to the mental health difficulties that these children experience. When children see their parents being torn away from them by border patrol agents, immigration judges, or prison guards, they witness those they love the most being humiliated, and taken away. These experiences likely have a profound and enduring impact that will last a lifetime because they watched those who made them feel safe become impotent when they were forcibly separated. Similarly, the later deportation of a parent can further lead the child to experience depression, insecurity, and loneliness [70]. Table 4 shows that there are worries and stressors both before and after migrating.

### 4.4. PHQ-8

Depression was also found to be disproportionately high within our sample, with routinely more than 20% of respondents on each question stating that they felt a depressive symptom at least some of the time in the last two weeks and, in other categories, much more frequently. In the last year, almost 40% of the respondents answered "yes" to the question, "Have you felt depressed or sad in the past year most days, even when you feel good sometimes?" These results support our key finding that one-third of our sample may be struggling with moderate to severe PTSD, as depression is a common PTSD comorbidity [62].

### 4.5. Child PTSD Symptoms Scale

One-third of respondents had moderate to severe symptoms of PTSD. Meanwhile, 12% of respondents reported no symptoms of PTSD as assessed by this scale. Those who reported no symptoms may actually have extremely intense symptoms that they cope with by disassociating. A dissociative state is caused by denial and avoidance and is part of the trauma sequelae of PTSD [62]. Part of surviving trauma is often forgetting [71]. Nevertheless, the results show how the lives of Central Americans cannot be reduced to possibly traumatic experiences. Some individuals may have no PTSD symptoms at all, many young immigrants may have some, and a few of them may have many symptoms, and they can be helped by adequate and accessible mental health services and a network of non-profits providing help to this population as they often do in the D.C. metropolitan region.

## 5. Limitations

The youth that participated in the study made it to the border and were placed with a sponsor. Therefore, our sample includes relatively privileged individuals in comparison with those who could not make or complete the trip, or the ones who were returned at the border. Because of our partial recruitment through afterschool programs, counselors, and social and legal service providers, our sample was privileged: interviewees were more likely to be enrolled in school than the overall Central American youth population in the Washington, D.C. metro area. Other qualitative findings in the study show that schools were instrumental to youth in developing friendships. Coupled with an *n* of 58, this probably understates the pressures faced by the larger population under study. However, our data about immigrant youth are insightful. Little is known about recent cohorts of unaccompanied Central American youth, and our survey data and instruments can be used for an individual-level analysis.

Given the self-reported format of the survey, these results may be underreporting negative mental health outcomes. Vocabulary related to mental health conditions used in the survey may have been unfamiliar to some of our respondents or understood differently in Spanish than in English [72]. There are cultural differences in interpretations and stigmas concerning mental health, which might have caused respondents to downplay, exaggerate, or altogether hide information or experiences [48,49]. Indeed, studies on mental health stigma and self-concealment have documented how some cultural elements in Latin communities make individuals wary of disclosing their mental health status or seeking out services [65,73–75]. Latin Americans with mental and/or psychosocial disabilities may experience employment discrimination; therefore, someone's willingness to conceal their mental health problems can impact their finances [76]. Furthermore, youth who had not thought or spoken openly about their feelings, especially those who were younger, may have misinterpreted or struggled to understand the survey questions.

Emblematic of possible underreporting, most of our respondents answered in the negative when asked if they struggled with "mental health problems." Nonetheless, many reported that they suffered from anxiety, depression, constant stress, or sadness. While negatively perceived emotions and stress cannot alone necessarily be defined as "mental health problems," they could be indicative of other mental health disorders [77–79]. These responses are especially important when considered alongside the finding from the Child PTSD Symptoms Scale that one-third of Central American minors may be struggling with moderate to severe PTSD. This grants us reason to believe that the actual values are much higher for both our sample and the Central American immigrant youth population at large.

Another limitation was that the answers regarding pre-migration circumstances were based on recollection rather than two different measurements in a longitudinal study. These findings further show the limits of self-reported physical and mental health data, particularly for people with little access to healthcare [80]. They also show the importance of conducting direct measurements during the study and of triangulating self-reported questions, validated scales, and open-ended questions, and the qualitative data that they generate.

The study respondents do not come from a probability random sample because this is a vulnerable and hard-to-reach population. There is no equivalent of a census, phonebook, or neighborhood to create a randomizing sample strategy: the study's recruitment strategy utilized non-profit organizations, law offices, and snowball sampling. Many respondents were found through schools, especially in Montgomery County. Thus, we do not have the stories of those who may have dropped out of school and who may report different experiences due to being embedded in the community in ways other than school. Another consideration is that the bulk of interviews were conducted during the first six months of 2017, namely, before the Trump administration enacted many of its immigration policies and practices, before they took effect, or before they came to public light. The 2016 election and the Trump administration's immigration policies and directives changed the conversation about Latino immigrants, Central America, and intercultural and community violence in several ways. Even so, our data provide a strong benchmark for how youth

fared before 2016. Finally, only a small proportion (15%) of the youth were undocumented, and legal papers are noted as a key determinant of whether an immigrant will integrate well. It is important to note that most UACs approach immigration authorities at the border and ask for asylum and family reunification due to their status as minors. This study only addresses those who were not sent back after turning themselves in to Customs and Border Protection. This study predates the Remain in Mexico program started by the Trump administration and its policy of family separation after detention or asylum application. It also predates the COVID 19 pandemic, which created higher economic and psychological stress for Central American immigrants. Therefore, the results presented here undoubtedly understate the situation of young migrants between 2018 and 2021.

Given the conditions in the countries of origin at the root of migration, the hard passage through Mexico and militarized international borders, and the criminalization of immigration and asylum-seeking, we assumed that all participants had exposure to many or at least one traumatic event during their lifetimes. We did not verify through a trauma exposure checklist because this could have been too upsetting to complete.

## 6. Conclusions: The Social Determinants of Mental Health

The youth we interviewed often perceived that they had little or no opportunities in their sending communities, whether due to the power and violence of gangs or the inadequacies of schooling and labor markets. Leaving the only home an individual has ever known, the notoriously dangerous journey to the United States, and the challenges of their circumstances upon arriving in the United States are all possible sources of PTSD symptoms [11].

While self-reported mental health measures are not iron-clad ways of understanding an individual's feelings, there is a clear trend. Each measure improved slightly after migration. Being in the United States improved mental well-being perceptions for some, but not all. In other words, mental health concerns did not worsen post-migration in any of the scales or questions used. For participants, emigration reduced some risks but brought many new challenges around familial and societal integration.

This paper shows the complex situations faced by immigrants after arrival in the U.S. [36]. Even if outcomes improved when coming to the U.S., these vignettes support our findings that a third of the respondents may be suffering from PTSD or depression. In line with the literature, we conclude that changes that allow for better access to social services, education, healthcare, and employment are needed to improve Central American youth and their families' mental health outcomes and prevent further trauma exposure. Mental health is not the simple result of individual-level experiences or pre-determined neurochemistry alone but is deeply affected by environmental factors that include poverty, inequality, and immigration policies.

**Author Contributions:** Conceptualization, E.C.; Data curation, E.C. and D.J.; Formal analysis, E.C., D.J. and S.F.; Funding acquisition, E.C. and E.H.; Investigation, E.C. and D.J.; Methodology, E.C., D.J., L.B. and E.H.; Project administration, E.C. and E.H.; Resources, E.C. and E.H.; Supervision, E.C. and E.H.; Validation, E.C., L.B. and D.J.; Visualization, E.C., D.J. and S.F.; Writing—original draft, E.C., D.J., J.C. and C.C.; Writing—review & editing, E.C., D.J., J.C., C.C., S.F., I.G., L.B., and E.H. All authors have read and agreed to the published version of the manuscript.

**Funding:** The results come from the project "Household Contexts and School Integration of Resettled Migrant Youth" administered by the Center for Latin American and Latino Studies (CLALS) in American University located in Washington, D.C. in 2017 directed by the co-Principal Investigators Ernesto Castañeda, Eric Hershberg, and Noemi Enchautegui de Jesus. A 2016-2017 Faculty Research Support Grant from the Provost's Office at American University provided the funds to conduct this project. The Center for Health Risk and Society at American University provided funding to Ernesto Castañeda, Daniel Jenks, and Carina Cione to work on this paper.

**Institutional Review Board Statement:** IRB approval was obtained from American University IRB-2016-227.

**Informed Consent Statement:** Audio-recorded verbal informed consent was obtained instead of written consent in order to protect the confidentiality of respondents, some of whom are applying for asylum and thus facing possible deportation if their applications were not approved.

**Acknowledgments:** The authors thank Dennis Stinchcomb for administering this project housed within CLALS. Noemi Enchautegui de Jesus, co-PI, helped design the study and carry out interviews. The team that conducted the interviews includes Aida Romero, Catie Prechtel, Natali Collazos, Cynthia Cristobal, Maria de Luna, and Ines Luengo de Krom. We also thank them and Cristian Mendoza Gomez and the other research assistants who helped transcribe, translate, enter, and code the interviews. We thank Kim Blankenship, Deanna L. Kerrigan, and Wendy Davis for their support. We thank the editors and three peer reviewers for the helpful queries, feedback, and support. All errors remain our own responsibility.

**Conflicts of Interest:** The authors declare no conflict of interest.

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
