# Peer review of "Symptoms of PTSD and Depression among Central American Immigrant Youth"

_traumacare, doi:10.3390/traumacare1020010_

Round 1
Reviewer 1 Report
The topic is of interest and applying it to a sample at risk of inclusion is important, but the study methodology and the description of the sample must be improved.The qualitative methodology used, for example in the collection and analysis of the opinions of the respondents, is poorly founded (“mental health problems”) 406 before or after migrating to the United States, with only 3 out of 58 responding “yes”) and its subsequent analysis lacks statistical basis. On the other hand, since it is such a specific sample, it should be described with greater precision to understand the results obtained: "The conducted a survey with 58 participants who were Central American immigrant 219 youth, 41 adults sponsoring recently arrived youth, and 23 social service providers in the 220 metropolitan Washington, D.C. regio". On the other hand, although the sample is disaggregated by sex, an analysis by gender is not made when mental health has a component associated with the social role of assigned gender. Finally, the article does not refer to the COVID pandemic, when there are already studies that show its influence on the mental health of the population, it is especially of the young and the underprivileged. Both components of the characteristics of the analyzed sample. So the data may not reflect current reality.
Reviewer 2 Report
Manuscript Review MDPI
Trauma Exposure and Mental Health of Central American Youth
This is a very important manuscript that has some serious issues that must be addressed. The discussion section lack important references to key statements and the IRB statement and interview questions and Informed Consent Form and author contribution MUSRT BE submitted for the review process due to ethical and federal regulatory concerns--this is not an option! The manuscript could benefit from tightening up as the entire paper tends to meander and wander all over the place--needs to be more succinct and concise. A statement about what type of Mixed Methods study this is must be included and a description of how the quantitative and qualitative data were synthesized and integrated according to Mixed Methods research basic principles. This is an important and valuable paper that has some very sloppy pieces to it and needs some expert editorial oversight to tighten it up
Below are more specific comments
Lines 14-15: Abstract should state that this is a Mixed Methods study using both quantitative and qualitative components, including survey data, directed interviews and validated questionnaires. Abstract should also state the type of Mixed Methods design they have chosen to use: Is this a sequential explanatory Mixed Methods design? Is this an exploratory Mixed Methods design? If the authors do not know and were unaware of this, they should at least do some background reading to understand that this study is clearly a Mixed Methods Study, which has a huge amount of background literature on how to conduct them properly and according to best practices, with proper Mixed Methods design, and how to properly take quantitative and qualitative data and do triangulation and synthesis of the results. It should also be mentioned in the abstract that directed qualitative interviews were conducted because that is a significant part of this paper.
Line 222-225: please also include if an advocate representing this multiply vulnerable population (youth; trauma; immigrant status) was also invited to sit on the IRB for this review, which should be a standard practice for vulnerable populations
Line 258: Replace “fminus” with “minus”
Line 260: replace “Spanish and is often by psychologists and clinicians internationally” with “Spanish and is often used by psychologists and clinicians internationally”
Line 267-277: Please specify whether the Child PTSD Symptom Scale (CPSS) has been validated in its Spanish translation form, as was stated for the PHQ-9 validated questionnaire.
Line 275: Replace “(Prácticamente en todo momento” with “(Prácticamente en todo momento)”
Line 285: Replace “killed in front of themselves” with “killed in front of them”
Line 315-316: Replace “were being were being processed” with “were being processed”
Line 322: Replace “El Salvador” with “from El Salvador”
For Table 2, why are there no statistics associated with these numbers? Please provide them and state whether they were significant or non-significant pre- and post-migration. As this is a quantitative part of this Mixed Methods study, it should have statistics associated with it
Line 424: Replace “Table 5 summarizes” with “Table 3 summarizes”
Table 3: In the open-ended questions, did the authors do an analysis of codes and themes, even by hand? It would be of value to know what the frequency of each code was, even if it was done by hand. Under the three categories listed (anxiety, sadness, constant stress), what were the most common codes?
Line 468: Replace “Many being found” with “Having found many of these individuals”
Line 476-478: Why are there no references I the Discussion section? This must be addressed. For example the statement: “There are cultural differences in interpretations and stigmas concerning mental health, which might have caused respondents to downplay, exaggerate, or altogether hide information or experiences,” should be referenced with actual research.
Line 484-486: Again, the statement “While these cannot alone necessarily be defined as “mental health problems, they can be indicative of other mental health issues or disorders,” needs to have references associated with it.
Lines 495-497: The statement “One potential reason for this discrepancy may be the widespread stigma around mental health around the globe. Additionally, knowing what their feelings mean is not always a clear or easy process for children and adolescents,” must be defended with references.
Line 499-500: Replace “had many challenges to face and confront” with “had many challenges to confront”
Line 504: Replace “in in the U.S.” with “in the U.S.”
Line 509: Replace “housing, food, and clothes” with “housing, food and clothes”
Line 516: Replace “since Carlos does not whether” with “since Carlos does not know whether”
Line 557: Replace “the reasons as to why their parents” with “the reasons why their parents”
Line 560: Replace “of emigrating” with “when emigrating”
Line 588-590: These statements need to be referenced: “Those who reported no symptoms may actually have extremely intense symptoms that they cope with by disassociating. A dissociative state is caused by denial and avoidance and is part of the trauma sequelae of PTSD.”
Line 599-601: Again, this statement needs to be referenced “These results support our key finding that one-third of our sample 599 may be struggling with moderate to severe PTSD, as depression is a common PTSD 600 comorbidity.”
Line 606-607: We do not know if this statement is statistically significant, so you must rephrase this, “In each measure, things improved slightly after migration.”
Line 633: Replace “treat mental health disorders, and prevent further trauma exposure.” with “to treat mental health disorders, and to prevent further trauma exposure.”
Line 635-642: This is extremely sloppy to not include these materials (Author Contributions; IRB statement, funding, informed consent statement, acknowledgements) for the primary review and is in fact not acceptable and very unprofessional. These materials need to be submitted as part of the primary review for ethical and scientific rigor and federal regulatory reasons.
Reviewer 3 Report
This manuscript describes mental health stressors of young immigrants and asylum seekers from Central America to the U.S. The risks of trauma before, during and after migration are presented as well as the impact of these traumas on mental health. Findings from the analysis of interviews and survey responses, including the PHQ-9 and CPSS scales from 58 youth immigrants in 2017 are presented. The descriptions of experiences by participants present a powerful depiction of possible causes of trauma. The survey and scale responses provide evidence of improvements to self-reported mental health after migration but ongoing presence of symptoms of depression, anxiety and PTSD. Limitations of the research are clearly described and taken into account.
My expertise does not lie with immigration related trauma, so this aspect of the research has not been assessed in depth as part of this review.
Overall, the manuscript presents important research and information and is appropriate for this journal. I recommend publishing this research following significant amendments to the manuscript as detailed below and then additional review.
Abstract:
Include that the research uses both qualitative and quantitative methods. The mixed-methods approach is a significant strength of the work and should be clearly started.
Introduction:
A lot of really helpful background information is presented, including through the use of very up-to-date research.
The aim of the research is not stated.
The introduction could be improved by focusing on the impact of immigration on youth mental health. For example, are all the examples given in lines 149-157 related to youth?
The general flow of the introduction could be improved (e.g. sentence ending at line 170).
Line 66: In text citation should have reference number in square brackets rather than the year of publication.
Lines 87-88: The word “and” should occur between items 7 and 8 instead of between items 6 and 7 of the list.
Lines 118-119: “Nonetheless, children left behind are more likely to feel abandoned and experience resentment wrongly.”. Consider instead “Nonetheless, children left behind are more likely to feel abandoned and resentful.”
Lines 124-126: “Once in the United States, immigrant children will still be at risk for mental health disorders due to poor living conditions, lack of opportunities, discrimination, limited access to federal resources, and fear of deportation and detention [16].”. Consider instead “Once in the United States, immigrant children face additional risks for mental health disorders due to poor living conditions, lack of opportunities, discrimination, limited access to federal resources, and fear of deportation and detention [16].”
The paragraph beginning on line 158 doesn’t seem relevant.
Lines 167-169: “This shows the mental health advantages of providing administrative avenues to immigrate and for family reunification.”. The evidence presented above pertains to more than just mental health. Consider changing this statement to reflect the broader advantages possible.
Line 172- paragraph on mental health among immigrants should also start with the statistics that is later presented on youth (Line 185 onwards).
Lines 205-213: Is all evidence presented here based on reference 39 or only the last sentence? Consider introducing the reference earlier if they are all reference 39. Otherwise, provide additional references from the earlier information.
Materials and Methods:
It is difficult to determine if appropriate methodology has been used as the research aim/question was not stated in the introduction. Based on inferences from the abstract regarding the research aim, the methodology used was appropriate.
The chronological sequence of the survey and repetition of mental and physical health status questions at different points of the survey is particularly commended.
The use of mixed methods is not currently clear from the methods. The results presented suggest that interviews were used in addition to the surveys and validated questionnaires, however there is no mention of interviews or qualitative methods in the methods section. How was the qualitative data collected and analysed?
Why has the ethics approval number not been included for peer review?
Line 231: Write out CPSS and PHQ-9 in full since it’s the first mention of it.
Line 258: Remove the letter “f” from before the word “minus”.
Lines 257-259: Provide a reference with evidence of PHQ-8 validation (current reference provides evidence of PHQ-9 validation).
Lines 259-260: Provide a reference with evidence of PHQ-8 validation in Spanish.
Table between lines 265 and 266: Display table number and title. Consider removing capitalisation of Depression on the final row (i.e., change to “Severe depression”) for consistency with the other rows in the table.
Line 275: Add closing parenthesis after “momento”.
Table 1: Add average age +/- SD. Should only use 1 decimal place.
Results:
The results provide examples of participants experiences from before, during and after migration.
Very important qualitative results were presented which was not expected based on the abstract, introduction, and methods.
Lines 284-286: “Youth were exposed to trauma or struggled with previous traumatic experiences. Youth shared that they had seen people, including loved ones, killed in front of themselves and personally experienced violence and abuse.” Please clarify if these experiences applied to all youth participants or a portion.
The quotes included are often very long. Only quotes from three participants are included. Consider including shorter quotes from more participants. Either way, the methods should include information about how qualitative data was collected, analysed and selected for the results.
Lines 315-316: Delete duplication of words “were being”.
Table 2 is a figure rather than a table, please correct this label. Additionally, if all respondents answered each of the questions for this figure, there is no need to show both yes and no responses. Instead, consider only showing yes response proportions, and use categories (colours) for pre- and post- migration. This will likely be better displayed with the axis flipped.
Line 424: Please check if the reference to table 5 should actually be to table 3.
Table 3: For consistency of capitalisation, in section about anxiety, consider starting the line “couldn’t afford school supplies” with a capital letter.
Lines 428-429: Some sections of the manuscript report use of the PHQ-8 but the heading for section 3.4 and the first line refer to the use of the PHQ-9. If the PHQ-8 was used, please consistently refer to that rather than the PHQ-9. This should also be consistent throughout the abstract, methods and discussion as well.
Line 433: Please check if the reference to table 3 should actually be to table 4.
Line 446: Was the clinician assessment part of the research?
Table 5 can be deleted and n and % can be included in the text.
Discussion:
Large sections of the discussion repeat information from the introduction but with fewer references. Other parts repeat what is already stated in the results. The first paragraph should summarise the main findings and then subsequent paragraphs should discuss the findings in more detail and compare to existing literature.
There is a body of work focusing on differences in the meaning of terms such as “mental health problems”, “mental illness”, etc. for different people, particularly for people of different cultural backgrounds. Consider including references to some of the existing literature on this matter in your discussion (approximately lines 491-497). An article that touches on the breadth and depth variation of these concepts which may be of use is https://doi.org/10.1080/10463283.2020.1796080.
Line 516: Consider adding the word “know” between “not” and “whether”.
Lines 519-529: This paragraph appears to present new results not previously included in the result section.
Lines 528-529 & 551-553: Information is stated as a result but is referenced as a concept (occurs twice). Please more clearly distinguish between results of your research and concepts from other research.
Line 534: A new term “sending communities” is introduced and used from here onwards. In earlier sections of the paper the term “country of origin” is used. Consider consistent use of terms throughout the manuscript.
Paragraph starting on line 603: Should be moved up since it’s a good summary of the mental health findings.
Other comments:
Could publication of this manuscript result in identification of participants who may be at risk of further harm from identification? Consider the relatively low number of participants within local areas and tight age ranges, use of first names and identification of a non-binary participant.
Author contributions, ethics approval details, funding and acknowledgements are required for peer review so should be included.
Lines 639: Consider rewording to “Verbal informed consent was obtained in order to protect the confidentiality of respondents applying for asylum and thus facing possible deportation.”
Reference 40 link does not work.
Author Response
Thank you for your feedback. Please see the document attached.

Round 2
Reviewer 1 Report
I believe the manuscript has been
sufficiently improved to warrant publication in Trauma Care.
Author Response
Thank you very much for your initial feedback, for reviewing this revision, and for supporting this project. Best wishes.
Reviewer 2 Report
The authors have addressed all of the concerns I had expressed and I feel the article is now suitable for publication in Trauma Care. This is a unique demographic with unique needs and a very important article that needs to be published to add to the existing literature in this area.
Author Response
Thank you very much for your initial feedback, for reviewing this revision, and for supporting this project. Best regards.
Reviewer 3 Report
The manuscript is very much improved from the original version and the authors have incorporated the suggested changes. Please see below some minor comments:
- Table 1: the first rows of the last column could be labelled with ‘years’.
- Figure 2 appears to be the first figure in the paper so should be figure 1. The paragraph above this figure refers to table 2 but should refer to this figure.
- Numbering of tables and figures throughout the paper need to be checked to ensure they are correct. In text references should be checked to ensure they match the corrected table/figure numbers. E.g. A new table 3 appears to have been added to the methods section which means the tables 3 and 4 in the results should now be tables 4 and 5.
- Table 4 title shouldn’t have “depression” capitalised.
- The cover letter states that table 5 will be deleted and results instead reported in text but this hasn’t been done.
Author Response
Dear Reviewer 3,
Thank you very much for your careful and useful initial feedback, for reviewing this revision and your further feedback and input, and for supporting this project overall.
- Table 1: the first rows of the last column could be labeled with ‘years’.
We have added years as well as % on the top of the right column. Thanks.
2. Figure 2 appears to be the first figure in the paper so should be figure 1. The paragraph above this figure refers to table 2 but should refer to this figure.
We have renamed this Figure 1 now.
- The numbering of tables and figures throughout the paper needs to be checked to ensure they are correct. In text references should be checked to ensure they match the corrected table/figure numbers. E.g. A new table 3 appears to have been added to the methods section which means the tables 3 and 4 in the results should now be tables 4 and 5.
Thank you for noticing, we have now triple-checked the naming, numbering, and ordering of the Tables and Figure.
3. Table 4 title shouldn’t have “depression” capitalised.
Done.
The cover letter states that table 5 will be deleted and results instead reported in the text but this hasn’t been done."
We ended up adding a new row to the table to report the results in a more nuanced way and thus decided to keep the table.
We did not make any substantive changes. But we gave the paper another round of co-author copy editing, and we changed the order of the sections in the discussion section to mirror the order presented in the results section, and moved all the discussions about limitations to the limitations section before the conclusion.
Thank you and best regards,
Castañeda et al.